# Improving Care Transitions for Hospitalized Veterans Discharged to Skilled Nursing Facilities: A Focus on Polypharmacy and Geriatric Syndromes

**DOI:** 10.3390/geriatrics4010019

**Published:** 2019-02-09

**Authors:** Amanda S. Mixon, Vivian M. Yeh, Sandra Simmons, James Powers, Eugene Wesley Ely, John Schnelle, Eduard E. Vasilevskis

**Affiliations:** 1Section of Hospital Medicine, Vanderbilt University Medical Center, Nashville, TN 37203, USA; ed.vasilevskis@vumc.org; 2Geriatric Research Education and Clinical Center (GRECC), VA Tennessee Valley Healthcare System, Nashville, TN 37212, USA; sandra.simmons@vumc.org (S.S.); james.powers@vumc.org (J.P.); wes.ely@vumc.org (E.W.E.); john.schnelle@Vanderbilt.Edu (J.S.); 3Center for Clinical Quality and Implementation Research, Vanderbilt University Medical Center, Nashville, TN 37203, USA; vivian.m.yeh@vumc.org; 4Center for Quality Aging, Vanderbilt University Medical Center, Nashville, TN 37203, USA; 5Division of Geriatrics, Vanderbilt University Medical Center, Nashville, TN 37232, USA

**Keywords:** polypharmacy, geriatric syndromes, care transitions, skilled nursing facilities, post-acute care, readmission

## Abstract

Geriatric syndromes and polypharmacy are common in older patients discharged to skilled nursing facilities (SNFs) and increase 30-day readmission risk. In a U.S.A. Department of Veterans Affairs (VA)-funded Quality Improvement study to improve care transitions from the VA hospital to area SNFs, Veterans (N = 134) were assessed for geriatric syndromes using standardized instruments as well as polypharmacy, defined as five or more medications. Warm handoffs were used to facilitate the transfer of this information. This paper describes the prevalence of geriatric syndromes, polypharmacy, and readmission rates. Veterans were prescribed an average of 14.7 medications at hospital discharge. Moreover, 75% of Veterans had more than two geriatric syndromes, some of which began during hospitalization. While this effort did not reduce 30-day readmissions, the high prevalence of geriatric syndromes and polypharmacy suggests that future efforts targeting these issues may be necessary to reduce readmissions among Veterans discharged to SNF.

## 1. Introduction

### 1.1. Problem Description

Older patients are at risk for unplanned hospital readmissions and the multitude of negative health outcomes associated with readmissions, including poor functional status and mortality [1,2,3,4,5]. Older adults discharged from an acute hospital to a skilled nursing facility (SNF) for short-term rehabilitation or nursing services are at especially high risk for readmission due to their diminished functional capacity, which may also heighten their sensitivity to the negative effects of geriatric syndromes and polypharmacy. One study found that patients discharged from the hospital to SNF to receive post-acute care (PAC) had a higher readmission risk (23.5%) than patients discharged directly to home [6]. Existing data on Department of Veterans Affairs (VA) readmission rates from SNFs suggest approximately a quarter of military Veterans are readmitted after hospital or SNF discharge [7,8]. The data from our local VA hospital was similar to national statistics, with a readmission rate of 22.7% indicating a need to improve care transitions for these vulnerable patients. 

### 1.2. Available Knowledge

Both geriatric syndromes and polypharmacy are common in hospitalized older adults and may increase their risk for readmission [9,10,11,12,13,14,15]. Geriatric syndromes are multifactorial, span multiple organ systems, and are associated with adverse outcomes [16]. Geriatric syndromes most commonly include: cognitive impairment, delirium, falls, depression, pain, incontinence, unintentional weight loss, and pressure ulcers. Polypharmacy (5 or more medications) and hyper-polypharmacy (10 or more medications) may also be considered geriatric syndromes. Approximately two-thirds of older adults have at least one of these geriatric syndromes present at hospital admission. Additionally, over the course of hospitalization, these syndromes often worsen in severity while new syndromes may also be acquired [17]. With the exception of falls and pressure ulcers, which are often the focus of hospital quality improvement efforts, the presence of geriatric syndromes is rarely communicated to the next provider at discharge [18]. Geriatric syndromes can weaken a patient’s functional status at hospital discharge, and a decline in mobility, self-care, and cognition have all been linked to an elevated risk of unplanned readmission within 30 days of hospital discharge [19,20,21]. Furthermore, the polypharmacy associated with multiple comorbidities common among older patients complicates medication regimens and increases the risk of poor medication adherence and medication errors, thereby increasing the risk of adverse drug events that contribute to readmissions [22,23,24,25,26]. During care transitions, such as hospital and SNF discharge, prescription medications are frequently changed (e.g., started, stopped, or changed dosages), further exposing patients to medication errors, medication-related adverse events, and readmissions [27,28]. In addition, one recent study showed that hospitalized older non-Veterans discharged to SNF are prescribed an average of six medications that may contribute to or exacerbate geriatric syndromes [29]. 

To understand the baseline frequencies of information communicated to SNF at hospital discharge, we audited 26 patient charts. Based on the elements suggested by practicing SNF clinicians, we found that few geriatric syndromes are documented in the discharge packet sent to SNF. Some examples of elements relevant to geriatric syndromes and the frequency of their absence from the discharge paperwork are as follows: pain in the hospital and how it was treated (65% of audited charts); cognitive status at discharge (39%); delirium at discharge (85%); depression and/or anxiety in the hospital (62%); mobility status at discharge (31%); urinary and fecal incontinence (46% and 65%, respectively); assistance required for feeding (65%); skin condition at discharge (62%).

### 1.3. Rationale and Aims

Ensuring the essential information in discharge packets are provided during the transition from acute care hospitals to SNFs may improve communication and the quality of the care transition for older Veterans, thereby reducing readmission risk. Our primary objective was to standardize the assessment of seven geriatric syndromes and polypharmacy and to communicate this information during care transition in a sample of Veterans discharged from a VA medical center to SNF. The syndromes were: cognitive impairment, delirium, probable depression, incontinence (bowel and/or bladder), unintentional weight loss, moderate to severe pain, history of falls, and pressure ulcers (stages 1–4). Standardized clinical information related to each of these geriatric syndromes was communicated to SNFs at the point of the care transition using the Nursing Transition Summary (NuTs) and the Medication Management Form (see Appendix A). By identifying the prevalence of geriatric syndromes and polypharmacy in this patient population, the long-term goal was to inform a bundled intervention to improve the quality of the care transition.

## 2. Materials and Methods

### 2.1. Study Setting, Design, and Context

We conducted a one-year quality improvement (QI) study to implement a bundle of interventions at a high complexity teaching hospital in the Department of Veterans Affairs. This paper followed the SQUIRE 2.0 reporting guidelines for QI studies [30]. First, we gathered institutional support and funding from the regional VA office prior to launching the study. We met with the service chiefs of medicine, surgery, nursing, and social work as their endorsement of the study was vital to obtaining buy in from frontline clinicians. For context, our tertiary care hospital located in the southeastern United States has 238 inpatient beds, staffed by six medicine teams, a geriatric evaluation and management team, and several surgical subspecialty teams (e.g., general surgery, orthopedics, cardiothoracic, vascular). There are approximately 8000 admissions annually and 600–700 referrals to SNFs.

Next, we assembled our QI team which consisted of a research nurse, advanced practice nurse (APN), pharmacists, geriatricians, gerontologists, hospitalists, social workers, and project coordinators. To administer the interventions, we had a dedicated patient care transitions advocate (TA). This role was shared by a research nurse and an APN, who were both trained on the use of the standardized geriatric syndrome assessments. During hospitalization, the TA interfaced daily with social work and case management about SNF referrals and dispositions, “coached” patients to become actively involved in the care transitions process, educated patients about the care received in SNFs, conducted goals of care and advanced care planning discussions, conducted geriatric assessments, apprised physicians daily of progress toward discharge to SNF, and solicited management plans from physicians. 

### 2.2. Participants

To identify eligible Veterans, the TA contacted the inpatient social workers during weekdays to determine which patients had been referred by physical or occupational therapists to a SNF and which patients had been admitted from a SNF to the hospital. We excluded patients admitted from long-term care, as they have different levels of functioning compared to patients being referred to SNF from the hospital.

Implementation of the geriatric syndrome assessments were part of a larger bundled intervention to improve care transitions called IMPACT (IMproving Post Acute Care Transitions), which we have implemented at another hospital [31]. IMPACT was approved as a QI study by the hospital’s Institutional Review Board who waived the requirement for informed written consent. Instead, trained research personnel provided a standardized description of the study to eligible Veterans, who had the right to refuse participation. If the Veteran was not able to participate due to mental status or impaired verbal communication, then a surrogate with permission to discuss health information could provide answers to some questions, such as the incontinence and nutrition assessment. 

### 2.3. Data Collection

Once the TA-approached patients were referred for SNF, they collected demographics (e.g., age, gender), the patient’s code status (e.g., presence of do not resuscitate order or request for limited interventions), as well as assessments of the geriatric syndromes described below. 

The Pain Assessment Interview was derived from the pain section of the Minimum Data Set (MDS), the screening tool that is completed in post-acute care within 72 h of a patient’s admission [32]. We chose to use this same set of questions to compare responses from acute care to post-acute care. The Pain Assessment Interview asks patients about pain management medications, the recent presence and frequency of pain, as well as the effect of pain on function. We asked patients to rate their current pain on a scale of 0 (no pain) to 10 (worst pain imaginable). Ratings 4 and above were categorized as moderate to severe pain.

The Brief Interview for Mental Status (BIMS) is also part of the MDS used in the post-acute setting [33,34]. The BIMS consists of seven items to assess attention, short-term recall, and temporal orientation, scored from 0 to 3 points. The total score ranges from 0 to 15, with higher scores reflecting less cognitive impairment (0–7: severe impairment; 8–12: moderate impairment; 13–15: cognitively intact).

The Brief Confusion Assessment Method (B-CAM) is a delirium screening tool [35]. The scoring is either positive or negative and depends on whether symptoms had acute onset or fluctuating course, the presence of inattention, an altered level of consciousness, and disorganized thinking. Patients are B-CAM positive, i.e. delirium present, if they have an altered mental status or fluctuating course and inattention, and either altered level of consciousness or disorganized thinking. Those not meeting the criteria are B-CAM negative.

To assess for depression, we utilized the Geriatric Depression Scale: Short Form (GDS: SF or GDS-5) [36,37]. There are five yes/no questions, and each yes answer counts 1 point. The total score ranges from 0 (no symptoms) to 5 (all symptoms present), where a GDS-5 score of 2 or more indicates probable depression. 

We initially planned to gather information related to incontinence status, body weight (upon hospital admission and discharge), and appetite (daily intake values) from the medical record and briefly confirmed this information with the patient during interview. However, we realized that this information was absent from the medical record for most patients, which necessitated that we ask patients and/or their surrogates directly for this information. Given the number of geriatric syndrome assessments included in this project and the overall patient response burden of all questions combined, we chose to ask just a few brief questions to determine the presence/absence of these conditions and whether it reflected a recent change, rather than adding a longer screening tool that also captured frequency, severity, and/or impact on other aspects of functioning. We acknowledge the limited utility of these screening assessments as a study limitation. Our current studies now include broader screening tools for both of these conditions based on our experience in this quality improvement effort. The incontinence assessment contained 2 parts (Appendix A). The first set of questions referred to any time during the patient’s hospitalization event, while the second set of questions was asked close to hospital discharge. Patients, caregivers, and nurses answered questions about the presence of bowel or bladder incontinence, independence with toileting, safety, and the presence and timing of incontinence episodes relative to hospitalization. 

Similar to incontinence, the nutrition assessment also contained questions to be asked at two different time points: during hospitalization and near discharge (Appendix A). The TA abstracted height and the most recent weight recorded in the medical record. Patients and caregivers were asked about unintentional weight loss in the prior month, changes in appetite, and the need for assistance with eating.

Additional data collected by the TA from the medical record included a history of falls in the prior three months and pressure ulcers present during hospitalization. These two items were documented on the nursing admission intake documentation for all inpatients as part of routine clinical care. 

### 2.4. Intervention

The geriatric assessments conducted in this study were part of a larger intervention to improve transition from the hospital to SNF. The principles comprising the intervention were drawn from a model for ideal transitions of care and are illustrated in Figure 1 [38].

#### 2.4.1. The Care Transitions Communication Tools

We developed and evaluated two standardized forms to communicate efficiently valuable, organized information to support safe care transitions: the Nursing Transition Summary (NuTS) and the Medication Management tool. The main purpose of these two forms was to send succinct, accurate information regarding a patient’s hospital stay, recommendations for care, and medication management to subsequent care providers.

The Nursing Transition Summary (NuTS) was completed by the TA, based on a combination of the standardized assessment tools to assess common geriatric syndromes discussed above and a review of the patient’s hospital medical record (Appendix A). Other items included were: functional status (ambulation, transfers, weight bearing status), fall risk, presence of a urinary catheter, diet, skin care needs, allergies, intravenous access, infection control concerns, respiratory status, assistive equipment needed, labs and radiological studies pending, follow up appointments, and important hospital contact information. The results of the assessments and medical record review were then summarized using the NuTS form to allow one summary document to be shared with multiple care providers post discharge. There is a specific section titled “highlights,” which delineates the key clinical issues of concern identified by the TA. These key clinical issues were highlighted both on the form and via telephone through a “warm hand-off” (see below). 

The second form was the Medication Management Form, which served a two-fold purpose: to serve as a reconciled discharge medication list/transfer orders and provide anticipatory guidance on medications for SNF staff. The Medication Management Form was completed by an inpatient pharmacist who reviewed the patient’s medical record for medications upon admission, medication changes during the hospital stay, and medication recommendations following discharge (Appendix A). Additionally, the pharmacist worked with the medical service teams to have medications changed appropriately prior to the patient’s hospital discharge. In particular, the pharmacist identified potentially inappropriate medications and followed up with the medical service team to determine if these medications could be discontinued prior to discharge. In the development of the form, SNF staff had requested that we include indications for medications, reasons for discontinuing medications in the hospital, guidance on duration of therapy, and last dose given of each medication. Based on best practices for medication reconciliation, we listed pre-admission medications followed by the action taken on each of those medications during hospitalization. The third column is the list of medications to be administered at the SNF and their indication. The comments column provides prospective management plans, such as titration schedules, monitoring plans, stop dates, and warnings for potential adverse events that may be problematic for a particular patient. This side-by-side comparison of the pre-admission medications and transfer orders facilitated a quick review of medications across settings of care. Additionally, specific instructions were provided about high-risk medications (e.g., warfarin). 

Our goal was to send both the NuTS and the Medication Management Form to the SNF prior to the patient’s discharge from the VA hospital; however, this was not always possible. For example, if a patient was discharged in the evening or on the weekend when the TA was not available, the two forms were sent electronically the next weekday morning. 

#### 2.4.2. Warm Handover

At the point of transfer from the hospital to the SNF, the TA attempted to contact a nurse at the SNF within 24 h for a warm handover. The purpose of the call was to ensure that the SNF staff received the NuTS and Medication Management Forms, answered questions related to the patient’s discharge status, and oriented the receiving provider to the “highlights” section in the NuTS, which were key concerns.

### 2.5. Measures

In addition to the geriatric syndrome measures described above, we abstracted the hospital length of stay and other patient demographic information from medical records. We recorded the total number of medications at discharge and noted the number of these medications associated with geriatric syndromes for a subset of 25 patients, which was a convenience sample. The criteria regarding association of medications with six geriatric syndromes was described previously [29]. We also monitored unplanned readmissions to the VA hospital within 30 days of discharge to SNF. All collected data were entered into the VA version of REDCap system [39]. 

### 2.6. Analysis

All continuous and categorical data were analyzed using descriptive statistics. Thirty-day unplanned readmission data were analyzed using a U-chart. A U-chart is a type of statistical process control chart, which plots count data with variable sample size versus time in months [40]. Thirty-day unplanned readmission rates for each month were calculated by dividing the number of unplanned readmissions by the monthly discharges to SNF.

## 3. Results

### 3.1. Patient Characteristics

Between August 2013 and July 2014, a total of 308 patients were referred to SNF and at least one geriatric syndrome assessment was completed for 134 (43.5%) of these patients. The primary reason that no assessments were conducted for the remaining 174 patients referred to SNF was that patients were discharged from the hospital before assessments could be initiated by the TA due to delays in notification of SNF referral to the TA. The 134 patients who completed at least one geriatric syndrome assessment still could have incomplete data due to inability or unwillingness to respond to interview questions or discharge prior to the completion of all assessments. Patient characteristics are reported in Table 1. The mean age of the 134 patients that comprised the sample for this QI study was 73.9 years ± standard deviation (SD) 10.2 years, 96.3% were male, and 85.7% were white. The mean and median hospital length of stay was 14.6 ± 10.3 and 11 days, respectively. Patients were discharged on an average of 14.7 medications (SD 5.3).

### 3.2. Geriatric Syndromes

The prevalence of geriatric syndromes for the sample of 134 patients for whom at least one geriatric syndrome assessment was completed is shown in Table 2. The majority of patients (62%) had a history of recent falls, while about 40% had cognitive impairment, probable depression, incontinence, and unintentional weight loss. The mean number of geriatric syndromes endorsed per patient was 2.6 ± 1.6, with 101 (75%) endorsing two or more syndromes and 58 (43%) endorsing three or more syndromes. 

### 3.3. Polypharmacy

Ninety-eight patients (80%) met criteria for hyper-polypharmacy (>10 medications) [41]. In the sub-sample of 25 patients for whom medications associated with geriatric syndromes were evaluated, all (100%) had at least one medication associated with geriatric syndromes, with a median of five syndrome-related medications per patient (mean 5.4 ± 2.4) (Table 2). 

### 3.4. Readmission Rate

In the year prior to our study, the baseline readmission rate was 22.7% for patients treated at our VA; however, this value includes all readmitted patients from all settings—not just SNFs. During our QI study from September 2013 to July 2014, the overall mean rate of readmissions from SNFs over the 11-month period was 24.8%, which was not significantly different from the baseline readmission rate (Figure 2). This number is too small to determine what patient level factors might have predicted these readmissions. 

## 4. Discussion

In this one-year QI study, we implemented standardized assessment of geriatric syndromes and medications as well as a warm handoff of this information to improve communication during care transitions from the hospital to SNFs for older Veterans. There was no significant change in readmission rate between baseline and the intervention period. However, we did note that the VA patients who participated in this study had a significant burden of geriatric syndromes, polypharmacy, and hyper-polypharmacy, which included many medications associated with geriatric syndromes. Prior to our study, geriatric syndromes were not being recognized, documented in full, and relayed to SNF staff; however, our study attempted to improve the exchange of this information. Moreover, our communication about polypharmacy to SNF was more than a list of transfer medications; rather, it provided more context and prospective guidance for SNF staff. Additionally, the tools developed from this study may contribute to safer transitions in this vulnerable population. 

In the context of prior investigations, current results underscore the importance of key elements to address during transitions of care from hospital to SNF. Previously, we conducted a similar study in the Medicare population discharged from an academic medical center to SNF. We noted that over 90% of patients had at least one geriatric syndrome, 55% had three or more, and all had polypharmacy, which is similar to the results found in this Veteran sample [18]. Moreover, when we followed these patients from hospital to SNF, we found that many of the geriatric syndromes remained present at SNF discharge (e.g., depression, loss of appetite, unintentional weight loss, moderate to severe pain, history of falls) [42]. Therefore, facilitating information transfer about the presence and severity of geriatric syndromes as well as providing prospective guidance on medications, especially those that may contribute to the development or exacerbate the severity of geriatric syndromes, may be critical elements to include in the hospital to SNF discharge packet.

Nonetheless, in this particular study, our efforts did not significantly impact the rate of hospital readmissions. This may have occurred for a number of reasons. In considering the implementation of another care transitions intervention, Project BOOST, Williams’s research group identified the “8 Ps” that increase risk for readmission [43]. While the BOOST intervention targeted patients returning home, several of these risk factors were also present in our patients going to SNF, namely: problems with medications (polypharmacy, high risk medications), physical limitations, poor social support, prior hospitalization, and the need for palliative care. The BOOST implementation guide underscores our findings about common issues in the hospital to SNF transition. We believe that an intervention to improve the transition from the hospital to SNF must address the multiple geriatric syndromes and the high number of medications. Efforts to reduce the number of medications, particularly medications associated with geriatric syndrome severity, should be of highest priority. Additionally, future studies that emphasize timely and effective communication between the care coordinators in each setting (hospital and SNF) to ensure proper receipt and use of discharge packets may yield more promising results [31].

### Study Limitations

First, we had a mean readmission rate from six of our hospital’s units to serve as a baseline rate; however, we did not have a specific rate for patients readmitted from SNFs. Second, there was a large number of patients discharged to SNF who could not be contacted in time to complete the assessments. There was not an efficient way at our VA hospital to alert research staff when patients were referred to SNF, as this information typically was not available in the medical record until shortly before discharge. The approach that finally resulted in earlier identification of SNF referrals required daily contact with each hospital social worker and the physical therapy team, which was effective but time-consuming. A more standardized method of noting in the electronic health record (EHR) when a referral to SNF is made would be helpful. In implementing such a method, clinicians may have more time to establish patient goals and expectations, perhaps having a meaningful impact on reducing readmissions. Third, we utilized brief questions to determine the presence/absence of weight loss and incontinence, rather than adding a longer screening tool. It may have been difficult to complete standardized tools, given the paucity of nursing documentation in the chart and the number of other geriatric syndrome assessments we were conducting. Fourth, we only tracked readmissions back to our hospital, thus, the readmission data may have been incomplete, and therefore possibly higher than observed. Fifth, we may not have seen a significant effect of our bundled intervention on the readmission rate because there are multitudes of reasons why patients are readmitted, which may not have been addressed in our intervention. We did not adjudicate reasons why patients were readmitted. Yet, from our prior work in a non-Veteran population, we know potentially avoidable readmissions may be as high as 30%. Patients, hospitalists, and SNF staff believed readmissions to be avoidable due to a variety of reasons: premature hospital discharge, poor discharge planning, unresolved and poorly managed clinical issues at the SNF, inadequate treatment at the SNF, improper medication management at the SNF, diagnostic issues, and poor decision-making regarding the transfer to SNF among reasons why these readmissions might have been preventable [31]. Without knowing the exact reasons for readmission, we suspect that our QI study reflects the challenges of improving care in a real world setting, with limitations in study personnel, data collection, and competing priorities. 

## 5. Conclusions

VA patients discharged from the hospital to SNF have multiple geriatric syndromes and polypharmacy, with many of the medications being associated with geriatric syndromes. Our results support the need for awareness of and attention to geriatric syndromes and polypharmacy during hospitalization, at care transitions, and at the SNF. With a focus on such issues, new models of care can be developed to lessen the morbidity from geriatric syndromes and polypharmacy while better supporting this population.

## Figures and Tables

**Figure 1 geriatrics-04-00019-f001:**
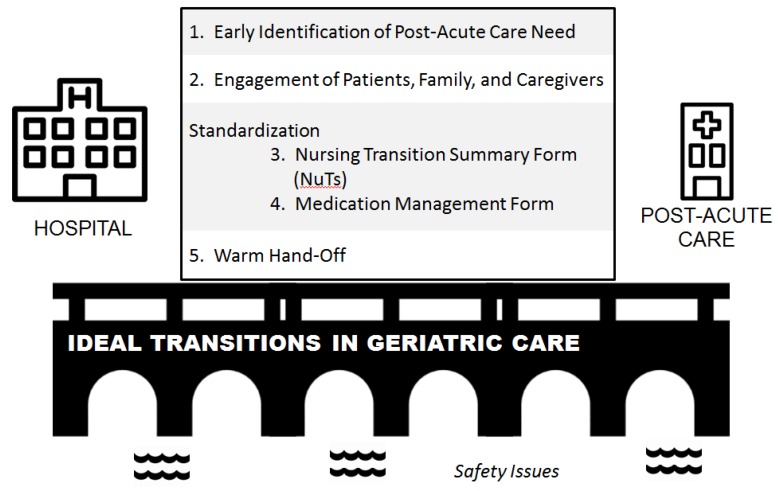
Essential elements of the IMPACT (IMproving Post Acute Care Transitions) program to ensure ideal transitions in geriatric care. Adapted from Burke et al. [38]. * Both the NuTS and the Medication Management Forms are reproduced in full in Appendix A.

**Figure 2 geriatrics-04-00019-f002:**
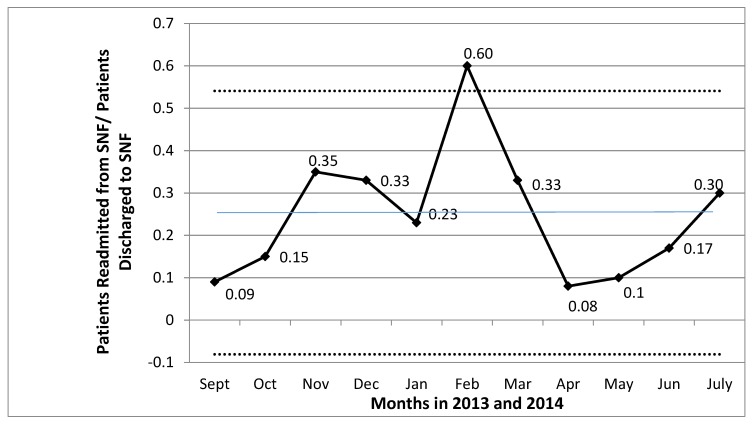
Statistical process control chart (U-chart) of unplanned 30-day readmission rates. Blue horizontal line indicates overall mean rate of readmissions of 24.8%, which was not statistically different from baseline/prior to the intervention. Dotted lines are two standard deviations from the mean.

**Table 1 geriatrics-04-00019-t001:** Patient characteristics.

Demographics	N = 134
Age, Mean ± SD	73.9 ± 10.2
Sex (Male)	96.3%
Race	
White	85.7%
Black	14.3%
Hospital length of stay in days	
Mean ± SD	14.6 ± 10.3
Median	11
Total number of medications at hospital discharge *, Mean ± SD	14.7 ± 5.3

* Includes pre-hospital, in-hospital, as needed, and routine.

**Table 2 geriatrics-04-00019-t002:** Prevalence of geriatric syndromes and medications associated with geriatric syndromes in Veterans Affairs (VA) patients discharged to skilled nursing facilities.

Geriatric Syndromes	Prevalence N = 134 ^1^	Medications Associated with Geriatric Syndromes (MAGS) ^2^ N = 25	
		Mean Number of MAGS ± SD	Proportion with ≥1 MAGS
Cognitive Impairment (Brief Interview for Mental Status, BIMS ≤ 12)	38.5% (45/117)	1.76 ± 1.13	88%
Delirium (Positive Brief Confusion Assessment Method, BCAM)	10.1% (13/129)	1.35 ± 1.11	84%
Probable Depression (Geriatric Depression Scale five-Item, GDS ≥ 2)	41.1% (44/107)	1.72 ± 1.34	80%
Incontinence (bowel and/or bladder)	39.6% (44/111)	1.57 ± 1.00	86%
Unintentional Weight Loss (in last 1 month or during hospitalization)	40.6% (28/69)	0.36 ± 0.64	28%
Moderate to Severe Pain (≥4 on a 0–10 rating scale)	23.7% (23/97)	N/A	N/A
History of Falls (in last 3 months)	61.7% (82/133)	5.08 ± 2.34	100%
Pressure Ulcers (at any point during hospitalization)	32.3% (43/133)	N/A	N/A

^1^ The total number of patients assessed for each syndrome varied due to refusal or patient was discharged prior to completion. ^2^ In this subsample, we examined which medications on a patient’s list were also on our published list of 513 medications associated with six geriatric syndromes: cognitive impairment, delirium, depression, urinary incontinence, reduced appetite or weight loss, and falls [29].

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
