# Peer review of "Improving Care Transitions for Hospitalized Veterans Discharged to Skilled Nursing Facilities: A Focus on Polypharmacy and Geriatric Syndromes"

_geriatrics, 2019, doi:10.3390/geriatrics4010019_

Reviewer 1 Report

No more comments

Author Response

We thank the reviewers for their comments.  There were no other comments from Reviewer 1. 

Reviewer 2 Report

Thank you for the opportunity to review the manuscript: "Improving Care Transitions for Hospitalized Veterans Discharged to Skilled Nursing Facilities: A Focus on Polypharmacy and Geriatric Syndromes."

The topic is significant for health care professionals in particular for personnel involved in Veterans care, and could have several consequences for people

The paper itself is written understandably and sounds appropriate, but some clarifications are needed.

The paper does not follow the author guidelines correctly for this kind of research, first of all, I want to suggest to redefine the article under the Journal Instruction for Authors.

References are aged, Authors proposed 32 references but, only 8 of them are since 2016 and a total of 12 since 2013. Probably a more in-deep recognition on what literature suggest about this topic could improve and update their research.

In particular:

Row 58: "polypharmacy... increase the risk of readmission" need a reference

row 80: "we audited 26 patient charts" were these enough? This question emerged because Authors have a respond ratio of 43,5% due to the lack of information, time or because patients refused to fulfil the assessment.

Row 167 and following: Authors decided to create two specific assessment tool to evaluate nutrition and incontinence. We can find in the literature many validated tools to do the same assessment and, for what concern the nutritional status, the appendix seems not fitted to assess the risk for patients. Can Authors explain their decision?

Row 223: can Authors explain if the limit of availability of the TA during the weekend had an impact on the study? if not, this paragraph is not necessary

Results:

as described before, patients characteristics could be improved to explain better why Authors were not able to perform all the geriatrics syndromes assessment; how many patients refused and why, for instance, can better tell us how do they cope with the assessment itself.

Authors declare they collect data for patients for whom at least one assessment was completed. Should we have more information about this? How many assessments performed for patients, which ones, maybe the can also describe why some evaluation was more easily achieved than others.

I also suggest a "limits of the study" section where authors can collect all the information they describe in the paper about Study limitation.

Author Response

1. The paper does not follow the author guidelines correctly for this kind of research, first of all, I want to suggest to redefine the article under the Journal Instruction for Authors.

--We utilized the template sent to us by Dr. Powers, who was a guest editor for the issue on  "Geriatric Care Models”. We believe this is a research manuscript with regard to type of publication, rather than a review or case report. In reviewing the Instructions for Authors, we have added full details of the authors’ affiliations. We have included the Introduction, Materials and Methods, Results, Discussion, and Conclusion. Additionally, we have included sub-sections within each heading that follows the SQUIRE 2.0 reporting guidelines for quality improvement (QI), which is appropriate for our study. (From the website squire-statement.org, “The SQUIRE guidelines provide a framework for reporting new knowledge about how to improve healthcare.”) We have made this clear in section 2.1.

2. References are aged, Authors proposed 32 references but, only 8 of them are since 2016 and a total of 12 since 2013. Probably a more in-deep recognition on what literature suggest about this topic could improve and update their research.

--We have included more recent references, with 14 of 43 from 2016 to the present.

3. Row 58: "polypharmacy... increase the risk of readmission" need a reference

--We have included references.

4. Row 80: "we audited 26 patient charts" were these enough? This question emerged because Authors have a respond ratio of 43,5% due to the lack of information, time or because patients refused to fulfil the assessment.

--We would like to clarify this point. The chart audits were conducted PRIOR to our intervention. The purpose was to understand what information was included in discharge paperwork of patients going from hospital to skilled nursing facilities (SNFs). We believe 26 charts was an adequate sample to identify deficits in discharge paperwork. It is not uncommon for QI work to begin with an assessment of the problem, which was the chart audit.

5. Row 167 and following: Authors decided to create two specific assessment tool to evaluate nutrition and incontinence. We can find in the literature many validated tools to do the same assessment and, for what concern the nutritional status, the appendix seems not fitted to assess the risk for patients. Can Authors explain their decision?

--We acknowledge that we did not use validated tools to evaluate nutrition and incontinence. Please see our explanation for the questions we did ask in section 2.3.

6. Row 223: can Authors explain if the limit of availability of the TA during the weekend had an impact on the study? if not, this paragraph is not necessary

--If the TA could not send the NuTS and Medication Management Forms prior to discharge, there would have been a delay in this information getting to the skilled nursing facility (SNF). However, the frequency of patient discharges in the evening or on the weekend was very low, so it is likely to have had a minimal impact on the study.

7. Results: As described before, patients characteristics could be improved to explain better why Authors were not able to perform all the geriatrics syndromes assessment; how many patients refused and why, for instance, can better tell us how do they cope with the assessment itself.

--In section 3.1 and Table 1, we have included the patient characteristics that we collected: “…a total of 308 patients were referred to SNF and at least one geriatric syndrome assessment was completed for 134 (43.5%) of these patients. The primary reason that no assessments were conducted for the remaining 174 patients referred to SNF was that patients were discharged from the hospital before assessments could be initiated by the TA due to delays in notification of SNF referral to the TA. The 134 patients who completed at least one geriatric syndrome assessment still could have incomplete data due to inability or unwillingness to respond to interview questions or discharge prior to the completion of all assessments. Patient characteristics are reported in Table 1.” We do not have information about why patients refused geriatric syndrome assessments.

8. Authors declare they collect data for patients for whom at least one assessment was completed. Should we have more information about this? How many assessments performed for patients, which ones, maybe the can also describe why some evaluation was more easily achieved than others.

--Table 2 shows the denominators for each geriatric syndrome, indicating how many patients answered the assessments. Of the geriatric syndromes, weight loss was assessed on only 69 patients. This information was difficult to ascertain from the medical record, so we did ask patients/caregivers about changes in weight, changes in appetite, and need for assistance with eating.

9. I also suggest a "limits of the study" section where authors can collect all the information they describe in the paper about Study limitation.

--We do have a limitations paragraph and have now clearly labeled it as section 4.1.

Round  2

Reviewer 2 Report

Thank you for giving me the opportunity to evaluate the revised paper. Authors have answered to all the questions and suggestions provided in the first review round, and I want to to thank them for their explanations. The paper is, in my opinion, now easy to read and understand.